# Vitamin D Deficiency in Chronic Childhood Disorders: Importance of Screening and Prevention

**DOI:** 10.3390/nu15122805

**Published:** 2023-06-19

**Authors:** Madhura Joshi, Suma Uday

**Affiliations:** 1Birmingham Women’s and Children’s Hospital, Steelhouse Lane, Birmingham B4 6NH, UK; madhura.joshi@nhs.net; 2Institute of Metabolism and Systems Research, University of Birmingham, Edgbaston B15 2TT, UK

**Keywords:** vitamin D, hypovitaminosis, chronic illness, screening, rickets, nutrition, osteomalacia, bone health, supplementation, treatment

## Abstract

Vitamin D plays a vital role in regulating calcium and phosphate metabolism and maintaining bone health. A state of prolonged or profound vitamin D deficiency (VDD) can result in rickets in children and osteomalacia in children and adults. Recent studies have demonstrated the pleiotropic action of vitamin D and identified its effects on multiple biological processes in addition to bone health. VDD is more prevalent in chronic childhood conditions such as long-standing systemic illnesses affecting the renal, liver, gastrointestinal, skin, neurologic and musculoskeletal systems. VDD superimposed on the underlying disease process and treatments that can adversely affect bone turnover can all add to the disease burden in these groups of children. The current review outlines the causes and mechanisms underlying poor bone health in certain groups of children and young people with chronic diseases with an emphasis on the proactive screening and treatment of VDD.

## 1. Introduction

Vitamin D is a fat-soluble seco-steroid hormone and is produced by all humans and animals [1,2]. Its role in regulating mineral homeostasis and bone health has been vastly studied [3]. The oldest reports of vitamin D deficiency (VDD) causing bony deformities date back to the 1600s [1]. We have come a long way in our understanding about its structure, functions, laboratory testing, commercial production and diseases related to deficiency states as well as their treatment [1,2,4,5,6,7,8,9]. Recent studies have explored the wider role of vitamin D, beyond the classic mineral homeostasis, affecting almost every organ system [10], with a particular focus on the immune system [11]. Optimising vitamin D is thought to have multiple health benefits in addition to the well-established effect of improving bone health [12]. Although the causal or regulatory role of vitamin D in some chronic childhood disorders is unclear, its role in maintaining bone health through mineral homeostasis is beyond doubt.

Cutaneous synthesis, using sunlight, is the predominant source of vitamin D in humans [13]. The ubiquity of sunlight is not enough to ensure vitamin D sufficiency and there is evidence for the increased prevalence of VDD even in regions with abundant sunshine [14,15]. Various environmental, socio-cultural and behavioural factors can predispose individuals to VDD. Common factors among these are decreased effective sunlight exposure due to excessive use of sun blocking creams, covered clothing, darker skin pigmentation, high-latitude residence or cloudy climate [16,17]. In addition to the above factors, chronic illnesses greatly increase the risk of VDD in affected individuals due to a multitude of reasons [10]. Several consensus guidelines have outlined principles for the screening and treatment of VDD in healthy at-risk children [18,19,20,21]. However, specific recommendations for the prevention and treatment of VDD in many chronic childhood diseases are lacking and general guidelines are often followed. This review aims to elucidate the role of vitamin D in chronic illnesses in childhood, with an emphasis on its skeletal effects and the need to ensure robust supplementation in specific high-risk cohorts.

## 2. Vitamin D Metabolism

Vitamin D exists in two natural forms, ergocalciferol (D2) synthesised by plants and fungi, and cholecalciferol (D3), by humans and animals. Ultraviolet-B (UV-B) rays from sunlight convert 7-dehydrocholesterol, a metabolite of cholesterol, to pre-vitamin D3, which is rapidly converted to cholecalciferol through a heat-induced isomerisation process in the skin [22]. Cholecalciferol is then transported to the liver, bound to albumin and vitamin-D-binding globulin (or vitamin D binding protein DBP). DBP, an alpha macroglobulin synthesised by the liver, is crucial for transporting cholecalciferol to the liver and kidneys for hydroxylation and conversion into active metabolites.

In the liver, 25-hydroxylase enzyme (CYP2R1) converts cholecalciferol and ergocalciferol to their 25-hydroxylated forms, 25-hydroxyvitamin D (25OHD), which is the most abundant circulating form. 25OHD undergoes further hydroxylation in the kidneys by 1 alpha hydroxylase enzyme (CYP27B1) to the active form (calcitriol or 1,25-dihydroxyvitamin D [1,25(OH)_2_D]) (Figure 1). Low calcium and phosphate levels and parathyroid hormone (PTH) enhance 1 alpha hydroxylase activity to produce calcitriol [23,24]. The primary action of calcitriol is on the intestinal mucosal cells to increase absorption of calcium and phosphate [24]. In these target cells, calcitriol binds to its receptor, commonly known as vitamin D receptor (VDR), and internalises [25]. This hormone receptor complex along with its co-receptor Retinoid-X-receptor then undergoes intranuclear translocation and binds to the vitamin-D-responsive element of the DNA, resulting in mRNA and protein synthesis. Calcitriol has a gene regulatory effect through which it enhances intestinal mucosal calcium absorption (by upregulation of calcium transporters) and maintains calcium homeostasis [22,25]. To limit the actions of vitamin D, the catabolic enzyme pathway comes into play wherein 24 hydroxylase enzyme (CYP24A1) converts 25OHD and 1,25(OH)_2_D to water-soluble excretable forms in the kidneys [26,27].

### Sources of Vitamin D

The most abundant source of cholecalciferol is from cutaneous synthesis on UV-B exposure (280–315 nm) from sunlight [13]. Dietary sources are limited with fatty fish, beef liver, egg yolk and dairy products being some of the sources. Given the limited dietary sources and multiple factors influencing the attainment of adequate vitamin D via UV-B exposure, reliance on supplements is a necessity in the absence of mandatory systematic food fortification [17].

## 3. Vitamin D Deficiency

### 3.1. Identifying the High-Risk Population

High-risk groups include healthy individuals with certain risk-factors for developing VDD and those with disease states that directly or indirectly affect vitamin D synthesis and/or metabolism (Table 1) [10].

The prevalence of VDD is closely linked to geographical locations such as latitude, weather conditions, ethnic characteristics, and cultural and social aspects [16]. The presence of more than one risk factor enhances the risk of VDD [28].

In the healthy high-risk individuals, the promotion of supplementation should be prioritised, and testing should only be offered to symptomatic individuals. In at-risk individuals with underlying health conditions, however, regular screening is recommended, for which circulating 25OHD serves as a reliable tool [20]. Routine screening of the general population is not recommended [20].

### 3.2. Biochemical Definition of Vitamin D Deficiency

The biochemical abnormalities in VDD precede its clinical and radiological manifestations [14]. Hence, determining an optimal level is crucial to guide treatment before the clinical syndrome of VDD sets in. The threshold for deficiency and sufficiency, advised by various organisations across the globe, differs. The thresholds for deficiency are elucidated based upon the inverse relationship of PTH and 25OHD as well as clinical signs [24]. The Endocrine Society defines 25OHD level > 75 nmol/L (30 ng/mL) to qualify for sufficiency and levels < 50 nmol/L (20 ng/mL) as deficient [20]. However, the Institute of Medicine (IOM, now National Academy of Medicine) definition [29], which is reiterated by the ‘Global consensus recommendations for prevention and treatment of nutritional rickets’ [18] includes:vitamin D sufficiency > 50 nmol/L (20 ng/mL);vitamin D insufficiency: 30–50 nmol/L (12–20 ng/mL);vitamin D deficiency < 30 nmol/L (12 ng/mL).

In addition to vitamin D, adequate dietary calcium intake (of 200–260 mg per day in infancy and 300–500 mg > 12 months of age) is recommended for prevention of nutritional rickets [14,18,29].

### 3.3. Pathophysiology of Vitamin-D-Deficient State

Low vitamin D levels initiate a cascade of physiological alterations that ultimately lead to the clinical syndrome of VDD (Figure 2). The most important of these are:Hypocalcaemia

VDD significantly decreases intestinal calcium absorption [25]. With a continued deficient state, blood calcium levels fall below a genetically predetermined threshold [30].

Secondary Hyperparathyroidism

Hypocalcaemia activates the calcium sensing receptors on the chief cells of the parathyroid glands, causing increased PTH secretion [24]. PTH enhances bone resorption and activates 1-alpha hydroxylase in the kidneys to synthesise calcitriol [31]. It also decreases renal calcium excretion to maintain normocalcaemia [32].

High Alkaline Phosphatase (ALP) or Hyperphosphatasaemia

High ALP, although not very specific, is an easily assessable marker of increased bone turn-over and should raise the suspicion of VDD if found to be raised in children [24,32]. It may be less helpful in conditions affecting the liver unless bone-and liver-specific isoenzymes are measured. An age-specific reference range should be used to interpret ALP values, as it is physiologically raised during periods of rapid growth in children [32].

Hypophosphataemia

PTH enhances renal phosphate excretion causing hypophosphatemia [33]. Ultimately, adequate phosphate levels are needed for the apoptosis of hypertrophic chondrocytes in the terminal layer of the growth plate in children. Normally, these apoptotic hypertrophic chondrocytes are replaced by osteoblasts, which lay down primary bone spongiosa along with concomitant vascular invasion [34]. The primary spongiosa is mineralised in the presence of adequate calcium and phosphate substrate. The hypophosphatemia-induced, continued accumulation of hypertrophic chondrocytes leads to growth plate widening and features of rickets [30,33].

Skeletal Hypomineralisation

The underlying pathophysiology in VDD is defective mineralisation of bone matrix (osteoid). When hypomineralisation occurs at epiphyseal growth plate cartilages and its adjoining metaphyseal region in children, it causes rickets, which is evident on radiographs [14]. In adults with VDD, during bone remodelling cycles, unmineralised osteoid is laid down in place of resorbed bone. The resultant soft bone is termed osteomalacia, which is diagnosed by histomorphometry analysis of bone biopsy specimens [35]. In children, both rickets and osteomalacia coexist. In chronic diseases, an interplay of various factors triggers the above pathophysiological alterations.

### 3.4. Clinical Syndrome of VDD

VDD either in isolation or together with dietary calcium deficiency causes calcipaenic rickets and osteomalacia [14,36]. Features of rickets depend on the age of presentation in addition to the severity and duration of VDD [36]. During periods of rapid growth (neonates, infants and adolescents) when the calcium requirement is high, VDD presents with symptoms of hypocalcaemia such as irritability, poor feeding, seizures in infants and tetany in adolescents [14]. In toddlers and children, in addition to hypocalcaemic features, classical signs of rickets such as frontal bossing, rachitic rosary, wrist and ankle swelling, bowing of the lower limbs and stunted growth may be evident [14,37]. Fractures can also be a presenting feature in VDD [28,38]. In children with underlying chronic conditions, the primary disease management is prioritised and less specific symptoms of VDD may go unnoticed until features of severe deficiency set in [39]. For instance, in metabolic bone disease associated with chronic kidney disease, biochemical abnormalities and bony features may not present until later in the disease process [40]. A detailed account of VDD in chronic childhood conditions is discussed below.

## 4. Vitamin D and Chronic Liver Disease

Bone health encompasses structural and functional skeletal integrity and, although it is genetically predetermined, it is closely linked to calcium and vitamin D homeostasis. Long-standing liver conditions are known to adversely affect bone health in children and adults [41]. Cirrhosis, non-cirrhotic biliary disease, autoimmune and viral hepatitis, infiltrative diseases such as haemochromatosis and Wilson’s disease, and liver transplant are all associated with varying but significant negative impacts on bone mass [41,42]. Skeletal manifestations of chronic liver disease (CLD) are collectively termed hepatic osteodystrophy (HO).

### 4.1. Pathophysiologic Mechanisms Involved in HO

Liver disease: Depending on the aetiopathogenesis of hepatic disease, the mechanism by which bone health is affected may vary. However, the final target is the bone remodelling unit (the site where remodelling occurs), resulting in an imbalance between bone formation and resorption [43]. Advanced liver disease has been associated with increased levels of sclerostin (secreted by osteocytes), which blocks the molecular signalling pathways necessary for osteoblast differentiation [44]. Autoimmune hepatitis and hepatitis C result in the release of cytokines and inflammatory factors (Interleukin 6, Interleukin 1b, Tumour necrosis factor alpha) having bone-resorptive action and, more importantly, via secretion of RANKL (receptor activator of nuclear factor kappa beta) by activated T-lymphocytes and fibroblasts, which activates osteoclasts [41,45]. Infiltrative and cholestatic diseases also negatively impact osteoblast differentiation and proliferation [46].Vitamin D deficiency: VDD is a key player in the development of HO. A higher prevalence of VDD in children and adults with chronic hepatic disease has been reported in the literature [47,48,49]. In a large cohort of patients with CLD (*n* = 118), Lee et al. observed that cirrhosis, African American race and female gender were independent risk determinants for severe VDD (25OHD < 17.5 nmol/L or 7 ng/mL) [49]. Decreased 25-hydroxylase activity, the malabsorption of fat-soluble vitamins, relatively insufficient vitamin D supplementation, malnutrition, low albumin and DBP, and poor sunlight exposure are some of the factors contributing to VDD in long-standing liver disease [50,51,52,53].Other factors: Associated growth failure and pubertal delay [54] may contribute to adverse bone health due to the absence of an anabolic effect of Insulin-like growth factor 1 (IGF-1) and sex hormones on bones [55]. Sarcopenia or low muscle mass and strength, due to impaired protein synthesis in CLD, contributes to low bone mass [53]. Steroid therapy for hepatic disease negatively impacts bone mass by enhancing the lifespan of existing osteoclasts, increasing the apoptosis of osteocytes and osteoblasts [56], supressing the formation of osteoblasts in the bone marrow and promoting loss of calcium through the kidneys and gut [57].

### 4.2. Biochemical and Clinical Manifestations

HO may manifest biochemically with classic features of vitamin D deficiency to include low 25OHD, raised serum PTH, normal or raised alkaline phosphatase and normal or low calcium/phosphate.

The clinical presentation in children can be with long bone fractures [58], vertebral compression fractures (VCFs) [59], low bone mineral density [49], radiological signs of rickets [49] and poor linear growth [55].

Worsening HO may manifest as severe VDD that is difficult to treat, prolonged low phosphate levels despite vitamin D therapy, persistently raised PTH or pathological fractures.

### 4.3. Recommendations for Monitoring and Treatment of VDD in CLD

There is a lack of consolidated paediatric guidelines on the prevention, monitoring and treatment of VDD in CLD. Given that CLD is a significant risk factor for VDD, it is strongly advised to assess 25OHD status. Serial bone profile (serum-adjusted calcium, phosphate, alkaline phosphatase) and 25OHD at diagnosis and at 3–6 monthly intervals, thereafter, combined with follow-up clinic visits is recommended [60]. More frequent monitoring may be warranted with worsening HO. Furthermore, due to the risk of vertebral fractures, children with long-standing liver disease should be screened with radiographs of the lateral spine at the time of liver transplant and at regular intervals thereafter [55,59,61].

The choice of preparation for the prevention and treatment of VDD is cholecalciferol or ergocalciferol and not vitamin D analogues (alfacalcidol or calcitriol), as the latter fails to replenish vitamin D stores [55]. Alfacalcidol or calcitriol add-on therapy may be beneficial in the acute management of severe VDD, for example, in children with hypocalcaemic seizures, to enhance intestinal calcium absorption. Such therapy, however, requires the careful monitoring of serum calcium and urinary calcium excretion, and its timely de-escalation with rising calcium levels is crucial to prevent hypercalcaemia and/or hypercalciuria with a long-term risk of nephrocalcinosis [62].

In healthy children (0–18 years), ‘The Endocrine Society’ recommends 400–600 IU of vitamin D2/D3 for routine supplementation and 2000 IU per day for at least 6 weeks, for treatment of VDD [20]. Higher doses of vitamin D (up to 3–10 times the usual) may be necessary for the prevention and treatment of VDD in children with liver failure with an aim to maintain 25OHD levels above 50 nmol/L [55,63]. Table 2 outlines the suggested doses for supplementation and treatment of VDD in CLD.

In children with CLD, if oral vitamin D is ineffective due to malabsorption or cannot be given (for example, in patients on total parenteral nutrition), treatment with intramuscular vitamin D (ergocalciferol) can be used to ensure increased bioavailability [55,63]. Ergocalciferol, owing to its long half-life, can be given as a single dose depending on the age and repeated monthly if necessary until vitamin D sufficiency is achieved [65]. The routine use of phosphate supplementation for correcting low serum phosphate levels in CLD is discouraged as this may lead to a further increase in PTH level and in turn worsen HO [33,66]. Correction of VDD restores normal serum phosphate levels.

## 5. Vitamin D Deficiency in Chronic Kidney Disease

The kidneys play a major role in maintaining calcium and phosphate homeostasis primarily by reabsorption of these minerals, thereby preventing their urinary loss, and by synthesis of the active form of vitamin D. Metabolic bone disease in chronic kidney disease (CKD) is termed renal osteodystrophy. It is multifactorial with VDD playing a significant role in its pathogenesis. Studies have shown a high prevalence of VDD in children with CKD (40–83%) as compared to healthy children [67,68,69].

### 5.1. Pathophysiology of Renal Osteodystrophy

In CKD, glomerular filtration, which is measured by creatinine clearance (Cr-cl), decreases progressively as the disease advances. A decline in Cr-cl determines the stage of CKD (stage 1–5) for therapeutic and prognostic purposes. Early on in CKD (stage 2 and 3), decreased phosphate excretion is sensed by the kidneys [40], which leads to activation of the FGF-23 (Fibroblast growth factor)–Klotho system in the kidneys to enhance phosphate excretion [70]. FGF-23 also down-regulates renal 1 alfa hydroxylase enzyme [70]. Furthermore, as the disease advances, there is less functional renal tissue available for calcitriol production, causing a decrease in its serum levels, thereby decreasing intestinal calcium absorption. Reduced calcium in turn triggers the secretion of PTH from the parathyroid glands that enhances bone resorption [69,71]. With advancing renal disease (stage 4), the glomerular filtration is severely affected, resulting in phosphate retention [69]. Stage 5 or end-stage renal disease is marked by low serum calcium levels, high phosphate and low calcitriol levels, which further increase PTH, causing secondary and, in chronic cases, tertiary hyperparathyroidism [72].

### 5.2. Biochemical and Clinical Manifestations

The biochemical features at various stages of renal disease are listed in Table 3. Often, biochemical abnormalities in calcium, phosphate and alkaline phosphatase levels may be incidentally picked up during ongoing monitoring for CKD [40]. The elevated PTH results in a high bone turnover state, causing changes in bone histomorphometry.

The acidotic state, hyperparathyroidism, dysregulated vitamin D and mineral metabolism lead to undermineralisation manifesting as rickets, osteomalacia, fractures, bone pain and poor linear growth [40,73]. The raised calcium phosphate product may cause calcifications in the blood vessels and soft tissues [73].

### 5.3. Recommendations for Monitoring and Treatment of VDD in CKD

Monitoring serum calcium, phosphate, ALP and PTH levels, every 6–12 months, 3–6 months and 1–3 months in stages 2 and 3, stage 4 and stage 5, respectively, is recommended by the ‘Kidney Disease: Improving Global Outcomes (KDIGO)’ and the ‘Kidney Disease Outcomes Quality Initiative (K/DOQI) guidelines’ [74,75]. It is recommended to check 25OHD at the first instance of reporting a raised PTH level that is above the acceptable target for the stage of CKD and, if PTH is within the normal range, then to monitor 25OHD annually [74,75]. Studies suggest accepting PTH levels within 1–2 times the upper limit of normal in stages 2–3 and 1.7–5 times the upper limit of normal in stage 4–5 of CKD [76]. Maintaining serum calcium levels within the normal range for age and the phosphate levels in the upper normal range for age is recommended [71,74].

Supplementation with vitamin D (ergocalciferol or cholecalciferol) to prevent VDD is routinely recommended in children with CKD [71,74,75]. The VDD treatment proposed by the European Society for Paediatric Nephrology and KDOQI (for children over 1 year of age) is detailed in Table 4 [75,77]. Optimal doses recommended for infants and neonates vary due to the lack of robust evidence; however, it is advised to use lower doses for treatment in this age group [75,77].

The above doses are a guide and, if serum calcium is high—for instance, in cases of tertiary hyperparathyroidism—one must consider lower doses of vitamin D replacement over a prolonged period of time to prevent exacerbation of hypercalcaemia. PTH-lowering measures are important in the management of renal osteodystrophy. Studies suggest accepting PTH levels within 1–2 times the upper limit of normal in stages 2–3 and 1.7–5 times the upper limit of normal in stage 4–5 of CKD [76]. Use of active vitamin D analogues, calcimimetic agents (which improve the sensitivity of calcium-sensing receptor to calcium in the parathyroid glands), or a combination of both may be needed to augment the PTH-lowering effect [74]. Whilst active vitamin D analogues may be essential given the underlying renal defect in conversion, they do not correct the coexistent VDD; hence, native forms of vitamin D (ergocalciferol/cholecalciferol) must be used in conjunction [78].

In advanced stages of CKD, children may require phosphate lowering-agents in addition to dietary phosphate restriction, the choice of which is based on calcium level [73]. It is essential to monitor growth and pubertal development in CKD [54] as these can have a direct bearing on optimal bone health. Growth hormone therapy has been recommended for optimising growth in this cohort [74,79].

## 6. Vitamin D and Chronic Malabsorptive Disorders

Children with chronic gastrointestinal (GI) illnesses are greatly predisposed to poor bone health due to multiple reasons [80]. Many conditions such as coeliac disease, inflammatory bowel disease, cystic fibrosis, pancreatitis and bypass surgeries directly affect the absorption of vitamin D and calcium, causing metabolic bone disease in children [81,82,83].

### 6.1. Pathophysiology of VDD in Chronic GI Disorders

Children with chronic GI conditions can be predisposed to VDD due to various factors.

Hampered absorption of vitamin D and minerals due to atrophic and or inflamed intestinal epithelium or reduced surface area for absorption in cases of gastric bypass surgery [84,85].

Insufficient dietary intake of calcium and vitamin D along with higher requirements.Disrupted enterohepatic circulation [85].Glucocorticoid use [56,57].Limited outdoor activities and sunlight exposure due to the chronic debilitating nature of the condition.

### 6.2. Biochemical and Clinical Manifestations

Biochemical features of VDD in GI disorders are similar to those seen in nutritional rickets as elaborated in previous sections. Children with chronic GI disorders have a propensity for low bone mineral density and metabolic bone disease [80,83,86,87]. Retarded linear growth as well as low muscle mass are prominent clinical manifestations of chronic GI diseases in children due to the underlying malnutrition and are known to contribute to the low bone mineral density [53,88]. Clinical manifestations may include long bone fractures following trivial trauma, vertebral compression fractures, rickets, osteomalacia, symptoms of hypocalcaemia and growth failure [89].

### 6.3. Recommendations for Prevention and Treatment of VDD in Chronic GI Disorders

There is a lack of specific recommendation pertaining to VDD in each of these disorders but, taking into consideration the high propensity of deficiency in chronic GI illnesses, the need for screening in this cohort is beyond doubt. Assessing serum 25OHD and bone profile at diagnosis along with continued monitoring at 6–12 month intervals, especially during colder months of the year, is recommended by the ‘British Society of Paediatric Gastroenterology Hepatology and Nutrition’ [82].

Although clear recommendations for optimal doses of vitamin D in children with chronic GI illness are lacking, most organisations advise routine vitamin D supplementation with 400 IU in infants, 600 IU in children and 600–1000 IU per day in adolescents to prevent VDD [18,20,90]. The ‘European society for Paediatric Gastroenterology Hepatology and Nutrition’ recommends maintaining 25OHD levels above 50 nmol/L in these children and advises the use of intramuscular preparations in cases of malabsorption or poor compliance to daily therapy [91]. For treatment of VDD in both children and adults with malabsorption, the Endocrine society recommends two or three times the normal doses (6000–10,000 IU per day) for 6–8 weeks and a subsequent maintenance dose of 3000 to 6000 IU daily [20]. In cases of poor compliance, 50,000 IU weekly given for a longer term may be used [20,90,91].

Cholecalciferol is thought to be better absorbed as compared to ergocalciferol and hence is the treatment of choice in these conditions [92]. However, a study (*n* = 71) from Boston reported that a six-week course of daily oral supplementation with 2000 IU of cholecalciferol is as effective as a weekly oral supplementation with 50,000 IU of ergocalciferol, in correcting VDD [93].

In addition to vitamin D, an adequate intake of calcium and other vitamins and minerals must be ascertained.

## 7. Vitamin D in Overweight and Obesity

Obesity levels in children and adolescents have risen to pandemic proportions in the last few decades [94]. It is postulated that obesity and VDD are closely associated [95]. Studies have observed a higher prevalence of VDD in obese children and adolescents [96,97,98]. The increased prevalence could possibly be because of common underlying risk factors such as poor outdoor physical activity shared by both conditions. Whether obesity is a consequence of or contributes to VDD, or if certain regulatory mechanisms exist between vitamin D activity and excess adiposity is yet to be clearly established [95]. It is nonetheless certain that there is a need to ensure robust vitamin D supplementation in this cohort.

### 7.1. Pathophysiology of VDD in Childhood Obesity

Factors thought to be responsible for VDD in obese individuals are listed below.

Sequestration and volumetric dilution effect: Increased deposition of vitamin D in adipose tissue makes it less bioavailable in obese individuals [99]. In simple terms, as compared to lean individuals, in obese individuals, vitamin D gets distributed in a larger volume of fat tissue, resulting in lower serum levels [100].Impaired metabolism: Obesity predisposes to non-alcoholic fatty liver [101]. Fat deposition in liver cells is postulated to cause decreased 25-hydroxylase activity as observed by Targher et al. in their study correlating hypovitaminosis D with the severity of hepatic steatosis [102].Behavioural effects: Sedentary lifestyle and low physical activity (including outdoors) are known environmental determinants of obesity [103], which also predispose to reduced sunlight exposure and low cutaneous vitamin D synthesis [16,104].Insufficient routine supplementation: The requirement of vitamin D in obese individuals is higher and often routine doses of supplementation or treatment may not ensure adequacy [105].

### 7.2. Clinical Manifestations

Childhood and adolescence are periods of dynamic changes of growth and peak bone mass accrual, respectively [106]. Obesity in this age predisposes to poor bone health and increases the risk of fractures [107]. Studies have shown that the changes in bone microarchitecture in obese children and adolescents increase their risk for long bone fractures [107,108]. A population-based cross-sectional study from electronic medical records (*n* = 913,178), of children aged 2–19 years, investigated the relationship between body mass index (BMI) for age category and risk for fractures and identified a higher odds ratio (OR) for lower-extremity fractures with increasing BMI [109]. Overweight (BMI ≥ 85th but <95th percentile for age), moderately obese (BMI ≥ 95th and <1.2 times 95th percentile for age) and extremely obese (BMI ≥ 1.2 times 95th percentile for age) patients all had an increased OR of fractures of the foot (OR, 1.14, 1.23 and 1.42, respectively), as well as those of the ankle, knee and leg (OR, 1.27, 1.28 and 1.51, respectively) [109].

In addition to fractures, children with obesity have other musculoskeletal problems due to the adverse effect of excess weight on the growth plate, such as slipped capital femoral epiphysis (causing hip pain and limp), Blount’s disease (disorder of tibial growth plate causing bowing of legs), and back, ankle and leg pain [110]. Concomitant VDD and its resultant adverse effects on the bone add to the skeletal problems arising from excess weight. A retrospective observational study on 890 obese children concluded that coexistent VDD increased the risk of Blount’s disease by 7.33% [111].

### 7.3. Recommendations for Prevention and Treatment of VDD in Childhood Obesity

Routine supplementation of all overweight and obese children is vital to prevent VDD. It is argued that the dose for routine supplementation and treatment in this group is higher compared to children with normal weight [105,112]. The Endocrine Society guidelines specify giving a 2–3 times higher dose in obese children and adolescents [20]. The supplementation of overweight and obese children aged 1–10 years with 1000 IU and those aged 11–18 years with 2000 IU of vitamin D2 or D3 orally, routinely, is advised [20,90,95,113].

Taking into consideration the increased demands of this cohort, it is suggested to treat VDD with 4000 IU per day (1–10 years) and 6000 IU per day (11–18 years) for 8 weeks followed by the daily supplementation doses for life with an aim to maintain 25OHD > 75 nmol/L [20].

## 8. Vitamin D and Chronic Neurologic and Myopathic Illnesses

Children with chronic neurological and neuro-muscular diseases are yet another vulnerable group predisposed to poor bone health [87,114]. Childhood epilepsy, cerebral palsy and muscular dystrophies such as Duchenne’s form a large cohort of children with increased fracture risk [115,116,117,118]. Factors contributing to VDD and poor bone health in this cohort include disease-specific factors such as limited ambulation and low muscle mass; treatment with bone-toxic drugs such as anticonvulsants or steroids; general factors related to chronic illnesses such as reduced sunlight exposure and poor nutrition [119]. Children with Duchenne Muscular Dystrophy (DMD) are prone to fractures due to osteoporosis secondary to muscular weakness and the gradual inability to weight-bear along with osteo-toxic steroid therapy. Systematic and proactive screening with a Dual-Energy X-ray Absorptiometry (DXA) scan and treatment with anti-resorptive agents are warranted in children with DMD [120]. Lateral spine radiographs or Lateral Vertebral Assessment (LVA) on DXA scans is useful in the assessment of vertebral fractures. DXA scans and LVA may not be feasible or clinically useful in children with severe scoliosis.

Children on a nasogastric tube or gastrostomy feeds may often receive formulas fortified with vitamin D, yet assessing for VDD remains vital to ensure the adequacy of supplementation. Bone profile and serum 25OHD should be monitored on a 6-monthly to annual basis in addition to the daily supplementation of vitamin D and minerals in age-appropriate doses [119,120]. The optimal dose of vitamin D in children with chronic neurologic and neuromuscular disorders is not vastly studied. Higher doses of vitamin D may be needed for routine supplementation and treatment of VDD [120], especially in children on antiepileptics and high-dose steroids (two to three times the recommended doses for children) [20].

## 9. Vitamin D and Chronic Skin Diseases

The prevalence of VDD in chronic dermatologic diseases such as psoriasis, ichthyosis, systemic lupus erythematosus, vitiligo, eczema and atopy are high compared to the general population, especially in children [121,122]. Poor bone health in paediatric chronic skin conditions could be due to a combination of disease-related, environmental and behavioural factors. For example, in autoimmune disorders, inflammatory cytokines are implicated in the imbalance between osteoblastic and osteoclastic activity, adversely affecting bone health [123]. Low levels of CYP27A1 and CYP27B1 in the keratinocytes in the skin lesions, particularly in psoriasis, are proposed to predispose to VDD [123,124]. Treatment with bone-toxic medications such as oral or potent topical steroids, calcineurin inhibitors and methotrexate are responsible for low bone mineral density in certain dermatological conditions [125,126,127,128]. Some of these disorders are accompanied with arthritis, limiting physical and outdoor activities and the resultant limited sunlight exposure [121,123]. Behavioural changes such as covering the skin due to the anticipation of flare-ups are common with cutaneous diseases that limit sunlight exposure [16] and potentiate VDD.

It is therefore important to identify these children as a distinct high-risk group for VDD and prevent it through appropriate medical advice on supplementation and actively screen when necessary to treat symptomatic individuals.

## 10. Conclusions

Children with chronic health conditions are predisposed to vitamin D deficiency (VDD) due to a myriad of factors that negatively impacts skeletal health, adding to the pre-existing morbidity. It is therefore imperative to proactively screen these children for VDD, at diagnosis and timely thereafter. Emphasising routine supplementation for life and treatment with higher doses as necessitated by the underlying disease should be ensured. Until discrete disease-specific guidelines for the optimisation of vitamin D status in children are available, clinicians should assess individual patients’ needs for vitamin D and ensure appropriate management in discussions with specialists. Practically, this can be achieved by incorporation of the vitamin D assessment, supplementation and treatment of its deficiency as an integral part of the disease management protocol. VDD as a preventable cause of poor bone health in chronic childhood illnesses warrants robust screening.

## Figures and Tables

**Figure 1 nutrients-15-02805-f001:**
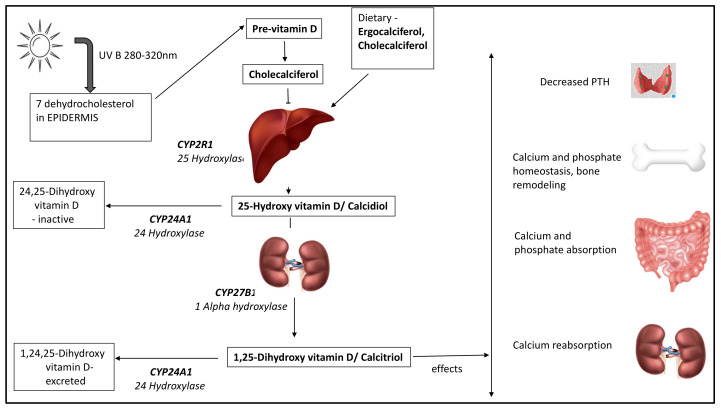
Vitamin D metabolism and its effects.

**Figure 2 nutrients-15-02805-f002:**
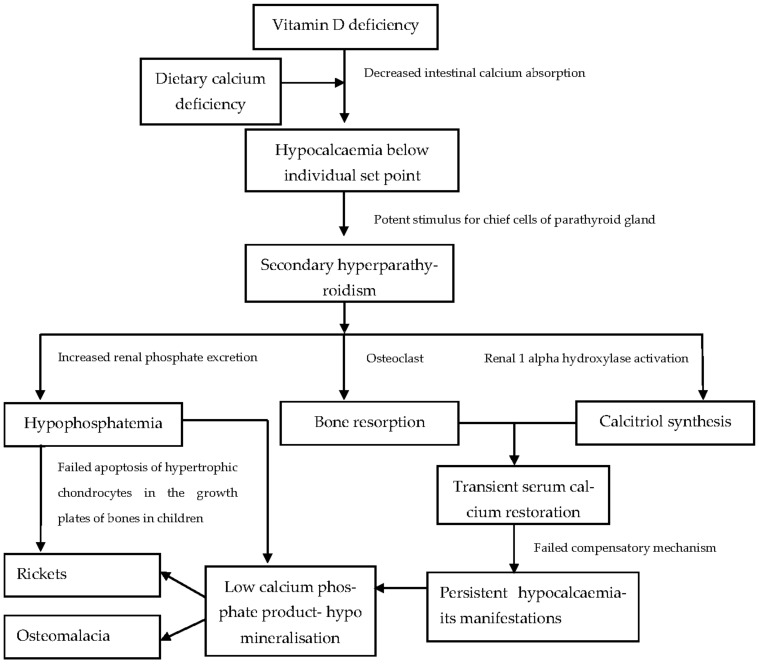
Pathophysiology of vitamin D deficiency.

**Table 1 nutrients-15-02805-t001:** Risk groups and the key factors for vitamin D deficiency.

Risk Group		Key Factors Responsible for Increased Risk of Vitamin D Deficiency
Living at high latitudesCloudy weatherPollutionInstitution or indoor dwellingExcessive use of sun-blocking creamsPigmented skinCovered clothing	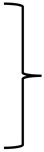	Lack and/or insufficiency of UV-Bexposure and/or penetration
Periods of growth spurts:Infancy, pubertyPregnancy	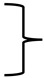	Periods of increased physiologicdemand
Aging		Reduced vitamin D synthesis
**Chronic illnesses**		
Chronic kidney disease		Decreased calcitriol synthesis
Chronic liver disease		Malabsorption, decreased calcitriol synthesis
Chronic Gastrointestinal disorders		Malabsorption
Chronic skin diseasesChronic neurologic illnesses	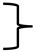	Poor UV-B exposure
**Obesity**		Reduced availability of vitamin D due to sequestration in fatty tissue
**Medications**		
Glucocorticoids		Induces catabolism of vitamin D
Anticonvulsants	
certain Antibiotics	

**Table 2 nutrients-15-02805-t002:** Suggested regimen for prevention and treatment of vitamin D deficiency (VDD) in chronic liver disease (CLD) [20,55,60,64].

Age (Years)	Daily Supplementation (Oral D2/D3 IU)	Treatment of VDD 12 Weeks (Oral D2/D3 IU)	I.M. ^1^ Single Dose (D2 IU)
0–1	2000	6000	50,000
>1–10	3000–4000	6000–10,000	150,000
11–18	4000–6000	10,000–12,000	300,000

^1^ IM (intramuscular) D2 (ergocalciferol) is indicated when VDD is not responding to oral therapy, due to malabsorption, and the I.M dose may need to be repeated monthly (for 2–3 months) until adequate serum 25OHD levels are achieved. D3-cholecalciferol, International Unit (IU).

**Table 3 nutrients-15-02805-t003:** Biochemical changes in chronic kidney disease (CKD) [40,70,72].

Stage of CKD	GFR (Cr-cl) mL/min	FGF-23	Calcium	Phosphate	Calcitriol	PTH
1	>90	N	N	N	N	N
2	60–89	↑	N	N	↓	N/↑
3	30–59	↑↑	N/↓	N/↑	↓↓	↑
4	15–29	↑↑	↓	↑↑	↓↓	↑↑
5	<15	↑↑	↓	↑↑	↓↓	↑↑↑

GFR—glomerular filtration rate, Cr-cl—creatinine clearance, FGF-23—fibroblast growth factor 23, PTH—parathyroid hormone, N—normal, ↑ increase, ↓ decrease.

**Table 4 nutrients-15-02805-t004:** Suggested regimen for prevention and treatment of VDD in CKD in children >1 year of age [71,74,75].

Stage of VDD in CKD	Treatment (Daily Dose Regimen) D2/D3	Treatment (Alternative Regimen) D2/D3	Follow-Up Supplementation D2/D3
Severe deficiency (<12 nmol/L or 5 ng/mL)	8000 IU for 1 month, then 4000 IU for 2 months	50,000 IU weekly for 1 month then 50,000 IU fortnightly for 2 months	After treatment- daily dose
Mild deficiency (12 to 50 nmol/L or 5–20 ng/mL)	4000 IU for 3 months	50,000 IU fortnightly for 3 months	0–1 year 400 IU 1–18 years 600 IU
Insufficiency (50 to 75 nmol/L or 20–30 ng/mL)	2000 IU for 3 months	50,000 IU once a month for 3 months	

VDD, vitamin D deficiency; CKD, chronic kidney disease; D2/D3, ergocalciferol/cholecalciferol; IU international units.

## Data Availability

Not applicable.

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
