# Peer review of "Vitamin D Deficiency in Chronic Childhood Disorders: Importance of Screening and Prevention"

_nutrients, 2023, doi:10.3390/nu15122805_

Round 1

Reviewer 1 Report

The topics of the review manuscript entitled "Vitamin D deficiency in chronic childhood disorders: Importance of screening and prevention" are very interesting and timely.

However, in general, I think that the authors should read again carefully and improve chapters 4. to 9. due to the fact that in places it is difficult to know whether they are describing the relationship "VDD and some disease entity" or vice versa. In this respect, the text should be precise and unambiguous. In addition, I think it should be made clear that the relationship is bidirectional, i.e. that VDD can accompany (or result from) the diseases in question and that it can contribute to the development or exacerbation of the course of these diseases.

In addition, a general observation is that the References section, although very extensive (it includes 129 items), only about 25% of the literature items are the most recent (dated within the last 5 years). In light of the fact that a very large number of manuscripts addressing the topic of vitamin D action have been published in recent years, I suggest that the proportion of more recent items in the References section should be increased.

I provide more detailed comments below:

- Please eliminate unnecessary colon symbols after section and subsection titles and in any unnecessary place. Please also eliminate unnecessary capital letters in table headings, figure captions and in section and subsection titles.

- Please place the captions of all figures below the figures and not above the figures - applies to Figures 1 and 3.

- Table 1: 1) I think it is too large, the formatting should be changed; 2) I suggest a minor change in the wording in the table heading: currently is "At risk groups and the key factors for vitamin D deficiency", I propose "Risk groups and the key factors for vitamin D deficiency"; 3) in the table currently is "Lack of UV-B exposure and/or penetration", I propose "Lack and/or insufficiency of UV-B exposure and/or penetration"; 4) the table currently has "Healthy at-risk group", I propose "Risk group"; 5) "Chronic liver disease" is a risk factor for VDD not only because of "Malabsorption" but also because of "Decreased calcitriol synthesis" (as the authors wrote in section 2. "Vitamin D Metabolism", the first step of vitamin D activation and DBP synthesis occurs in the liver) - please complete this section of the table; 6) I find the listing of rifampicin among the drugs that induce vitamin D catabolism problematic, as it is a strictly specific drug that is not necessarily known to every reader. In the case of "Medications", it would rather be written that the table gives examples of groups of drugs with the stated effect and not give their specific names.

- Lines 108-110 - please state what the text refers to and the source item.

- Figure 2: 1) the formatting of this figure should be corrected as in several places the inserted arrows are not in the right place; 2) please eliminate unnecessary capital letters and remove the abbreviation "Sr." - please insert the full name.

- Figure 3: I do not really understand the idea or the message of this figure. In my opinion the content of this figure does not reflect its caption. Reading the relevant part of the text goes a long way to understanding the content of the figure, but I believe that any figure should be readable without having to read the text.

- Please insert an explanation of all abbreviations and symbols in Table 3 and 4.

- Lines 390-393 in subsection 7.1 "Pathophysiology of VDD in childhood obesity": the authors state that among the behavioural factors predisposing to VDD in childhood is "staying indoors (increased screen time)". - I believe that this is not a factor that strictly corresponds to overweight or obese children. It is a more common factor and to attribute it to just this group of people I think is incorrect. I would suggest specifying this reason to 'low physical activity (including outdoors)'.

- Section 10, "Conclusion", provides a very apt summary of the content of the manuscript, while I believe this section should be much shorter.

- I ask for a careful adaptation of the "References" section to the current editorial requirements of the journal.

- Please standardise the writing of "vitamin D" throughout the text, as in some places it is "vitamin D" and in others "Vitamin D". Additionally, I do not think it makes sense to write the word 'vitamin' with a capital letter.

In conclusion, I think that the manuscript is very interesting and valuable in terms of content and, with all the above comments taken into account, it could be published in a journal.

Author Response

The topics of the review manuscript entitled "Vitamin D deficiency in chronic childhood disorders: Importance of screening and prevention" are very interesting and timely. 

However, in general, I think that the authors should read again carefully and improve chapters 4. to 9. due to the fact that in places it is difficult to know whether they are describing the relationship "VDD and some disease entity" or vice versa. In this respect, the text should be precise and unambiguous. In addition, I think it should be made clear that the relationship is bidirectional, i.e. that VDD can accompany (or result from) the diseases in question and that it can contribute to the development or exacerbation of the course of these diseases. 

Response: Thank you for your feedback. Given that the causative role of vitamin D deficiency in the conditions we report is not yet established we have opted to restrict our report and advice to the proven skeletal manifestations of vitamin D deficiency in chronic disorders. We had previously stated this in our conclusion section, however, to provide clarity in response to your comment we have now moved this to the introduction paragraph.

Changes: Line 32-34

In addition, a general observation is that the References section, although very extensive (it includes 129 items), only about 25% of the literature items are the most recent (dated within the last 5 years). In light of the fact that a very large number of manuscripts addressing the topic of vitamin D action have been published in recent years, I suggest that the proportion of more recent items in the References section should be increased.

Response: We have included most recent guidelines and recommendations available for the diagnosis and management of vitamin D deficiency for specific disorders discussed and also the most up to date global consensus guidelines for prevention and management of nutritional rickets and osteomalacia. Unfortunately, there is a paucity of recent literature in certain conditions such as liver diseases in children. However, if there are any specific recent references that you feel have been missed and suggest including, we would be happy to amend. In response to the other reviewer comment we have additionally highlighted the references deemed to be of importance.

I provide more detailed comments below:

- Please eliminate unnecessary colon symbols after section and subsection titles and in any unnecessary place. Please also eliminate unnecessary capital letters in table headings, figure captions and in section and subsection titles.

Response: We have now included these changes in the indicated sections.

- Please place the captions of all figures below the figures and not above the figures - applies to Figures 1 and 3.

Response:  We have made the above amendment to figure 1. We have excluded figure 3 in response to one of your comments.

- Table 1: 1) I think it is too large, the formatting should be changed; 2) I suggest a minor change in the wording in the table heading: currently is "At risk groups and the key factors for vitamin D deficiency", I propose "Risk groups and the key factors for vitamin D deficiency"; 3) in the table currently is "Lack of UV-B exposure and/or penetration", I propose "Lack and/or insufficiency of UV-B exposure and/or penetration"; 4) the table currently has "Healthy at-risk group", I propose "Risk group"; 5) "Chronic liver disease" is a risk factor for VDD not only because of "Malabsorption" but also because of "Decreased calcitriol synthesis" (as the authors wrote in section 2. "Vitamin D Metabolism", the first step of vitamin D activation and DBP synthesis occurs in the liver) - please complete this section of the table; 6) I find the listing of rifampicin among the drugs that induce vitamin D catabolism problematic, as it is a strictly specific drug that is not necessarily known to every reader. In the case of "Medications", it would rather be written that the table gives examples of groups of drugs with the stated effect and not give their specific names.

Response: Thank you for the suggestions. We have incorporated all the helpful clinical suggestions made. The formatting has unfortunately been changed by the journal following our submission. We hope to be able to address any formatting issues prior to publication.

- Lines 108-110 - please state what the text refers to and the source item. 

Response: We have added vitamin D as the prefix. The source item is clarified just above the classification: line 109 - ‘Global consensus recommendations for prevention and treatment of nutritional rickets’ (18).

- Figure 2: 1) the formatting of this figure should be corrected as in several places the inserted arrows are not in the right place; 2) please eliminate unnecessary capital letters and remove the abbreviation "Sr." - please insert the full name.

Response: We agree and have incorporated the changes suggested in Figure 2.

- Figure 3: I do not really understand the idea or the message of this figure. In my opinion the content of this figure does not reflect its caption. Reading the relevant part of the text goes a long way to understanding the content of the figure, but I believe that any figure should be readable without having to read the text.

Response: We agree that figure 3 is not adding any additional value and have now excluded it.

- Please insert an explanation of all abbreviations and symbols in Table 3 and 4.

Response: We have explained all abbreviations and symbols in Table 3 and 4.

- Lines 390-393 in subsection 7.1 "Pathophysiology of VDD in childhood obesity": the authors state that among the behavioural factors predisposing to VDD in childhood is "staying indoors (increased screen time)". - I believe that this is not a factor that strictly corresponds to overweight or obese children. It is a more common factor and to attribute it to just this group of people I think is incorrect. I would suggest specifying this reason to 'low physical activity (including outdoors)'.

Response: Thank you for the feedback. We have amended the sentence to include ‘low physical activity (including outdoors)’ in place of ‘staying indoors (increased screen time)’.

- Section 10, "Conclusion", provides a very apt summary of the content of the manuscript, while I believe this section should be much shorter.

Response: We have now made it more concise.

- I ask for a careful adaptation of the "References" section to the current editorial requirements of the journal.

Response: The referencing style used is in keeping with the instructions we were provided. However, we would be happy to adapt as per any change in requirement.

- Please standardise the writing of "vitamin D" throughout the text, as in some places it is "vitamin D" and in others "Vitamin D". Additionally, I do not think it makes sense to write the word 'vitamin' with a capital letter.

Response: We agree and have made the above amendments.

In conclusion, I think that the manuscript is very interesting and valuable in terms of content and, with all the above comments taken into account, it could be published in a journal.

Response: Thank you for the positive feedback.

Reviewer 2 Report

The study presents a comprehensive review of the role of vitamin D and what its deficiency can cause in children and young people. The study presents clear language and well-defined methodological procedures.

The study presents a comprehensive review of the role of vitamin D and what its deficiency can cause in children and young people.

Introduction: presents a clear explanation of the objectives of the study;

Methodology: clear and objective for the proposal of a broad literature review;

Results and discussion: high point of the study, very well founded, addressing conceptual aspects of vitamin D, biochemical and pathophysiological relationships of its deficiency in childhood and adolescence.

Conclusions: consistent with the objective of the study.

In addition, it is worth highlighting the references used, which are up-to-date and robust.

Author Response

The study presents a comprehensive review of the role of vitamin D and what its deficiency can cause in children and young people. The study presents clear language and well-defined methodological procedures.

The study presents a comprehensive review of the role of vitamin D and what its deficiency can cause in children and young people.

Introduction: presents a clear explanation of the objectives of the study;

Methodology: clear and objective for the proposal of a broad literature review;

Results and discussion: high point of the study, very well founded, addressing conceptual aspects of vitamin D, biochemical and pathophysiological relationships of its deficiency in childhood and adolescence.

Conclusions: consistent with the objective of the study.

In addition, it is worth highlighting the references used, which are up-to-date and robust.

Response: Thank you very much for the positive feedback and also for the very helpful suggestion of highlighting key refences which we have now incorporated.
